# Accessible luminal interface of bovine rectal organoids generated from cryopreserved biopsy tissues

**Minae Kawasaki**, **Yoko M. Ambrosini** *

Department of Veterinary Clinical Sciences, College of Veterinary Medicine, Washington State University, Pullman, Washington, United States of America

* yoko.ambrosini@wsu.edu

## Abstract

Developing precise species-specific *in vitro* models that closely resemble *in vivo* intestinal tissues is essential for advancing our understanding of gastrointestinal physiology and associated diseases. This is especially crucial in examining host-pathogen interactions, particularly in bovines, a known reservoir for microbes and pathogens posing substantial public health threats. This research investigated the viability of producing bovine rectal organoids from cryopreserved tissues. We compared two cryopreservation methods with a traditional technique using fresh tissues, evaluating their effectiveness through growth rates, long-term viability, and comprehensive structural, cellular, and genetic analyses. These assessments utilized phase-contrast imaging, immunofluorescence imaging, and RT-qPCR assays. Additionally, the study developed a sophisticated method for forming a functional epithelial barrier from organoid-derived bovine rectal monolayers, incorporating a wide range of epithelial cells. This methodology employed transepithelial electrical resistance (TEER), parallel artificial membrane permeability assay ($P_{app}$), confocal microscopy, and advanced imaging techniques like scanning electron microscopy (SEM) and transmission electron microscopy (TEM). Our findings decisively show that bovine rectal organoids can be effectively generated from cryopreserved biopsy tissues. Moreover, we formulated a robust and optimized protocol for creating functional rectal monolayers from these organoids. This significant progress is particularly relevant given the susceptibility of the bovine rectum to various enteric pathogens of public health concern, marking a vital step forward in veterinary and biomedical research. The creation of accurate species specific *in vitro* models that faithfully mimic *in vivo* intestinal tissues is critical for enhancing our understanding of gut physiology and related pathologies. This is particularly relevant in studying the interactions between hosts and microbes or pathogens with significant public health risks where bovine can be the major reservoir.

## Introduction

Intestinal epithelial cells are fundamental for animal survival and growth, performing critical functions such as nutrient absorption and protection against pathogenic and toxicological

**Data Availability Statement:** All relevant data are within the manuscript and its Supporting Information files.

**Funding:** This study was supported in part by the Office of the Director National Institutes of Health

(K01OD030515 and R21OD031903 to YMA; https://www.nih.gov/institutes-nih/nih-office-director). The funders had no role in study design, data collection and analysis, decision to publish, or preparation of the manuscript.

**Competing interests:** The authors have declared that no competing interests exist.

threats from the external environment [1,2]. The development of accurate *in vitro* models that replicate intestinal tissues *in vivo* is essential for advancing our understanding of normal gut physiology and pathologies. This includes studying interactions between host and microbes or pathogens, and immune responses to intestinal infections like enterohemorrhagic *Escherichia coli* (EHEC) and *Salmonella enterica*, which pose significant public health concerns [3].

The field of study has been transformed by the advancement of organoid technology. This technology offers enhanced *in vitro* models compared to traditional cell cultures, including three-dimensional (3D) intestinal organoids [4], 2D monolayers derived from organoids [5], and sophisticated microfluidic intestine-on-a-chip systems [6]. While there is extensive literature on murine and human intestinal organoids and associated technologies [7–11], research on bovine equivalents is still emerging [12–14].

The requirement for fresh, viable tissue samples in generating organoids from adult stem cells significantly hinders the use of patient-derived tissues in research [15]. This issue is particularly pronounced in farm animals, where sample collection sites are often distant from laboratories. To address this, some studies have explored generating organoids from frozen human tissues or frozen intestinal crypts in horses and pigs [15–18]. However, similar approaches using bovine intestinal tissues have not been reported.

In cattle, the terminal rectum is a primary site for EHEC colonization [19–21]. While cattle usually show no clinical symptoms, EHEC can cause severe illnesses in humans [21]. Cattle are considered a major reservoir for EHEC, posing a risk for human infection [19–20]. Other pathogens such as *Salmonella spp.*, *Streptococcus agalactiae*, and *Campylobacter laminae*, which can potentially transmit to humans, have also been found in bovine rectal colonization [3,22,23]. Therefore, understanding host-pathogen interactions in the bovine rectum is crucial for controlling pathogen colonization in cattle and preventing human zoonotic infections.

This study evaluated the feasibility of producing organoids from cryopreserved bovine rectal tissues by comparing two cryopreservation methods against fresh tissue techniques. The effectiveness of these methods in generating bovine rectal organoids was assessed by comparing growth rates, long-term viability, and structural, cellular, and genetic profiles of organoids derived from frozen and fresh tissues. Additionally, this study introduced a robust method for creating a functional epithelial barrier from organoid-derived bovine rectal monolayers, comprising a diverse population of epithelial cells.

## Materials and methods

### Tissue sampling and cryopreservation

Rectal tissue samples were obtained from three cattle, aged between 15 and 18 months, at a local slaughterhouse using biopsy forceps (Hildyard Post-Nasal Biopsy Forceps, Med-Plus) as described previously [4]. Each animal yielded about 30 tissue pieces from a randomly selected area of the rectum, which were promptly immersed in a 50 mL conical tube filled with an ice-cooled wash medium composed of Dulbecco's phosphate-buffered saline (PBS) (Gibco) supplemented with 1x Penicillin/Streptomycin (Gibco) and 25 µg/mL Gentamicin (Gibco). The samples were kept on ice until further processing at the lab.

In the lab, the samples underwent a sterile wash with the wash medium described above and were segregated into three equal portions (around 10 pieces each), designated as "fresh tissue (FT)", "slow-frozen tissue (SFT)" and "flash-frozen tissue (FFT)". The "FT" samples were immediately treated with a 20 mM ethylenediaminetetraacetic acid (EDTA) solution for intestinal crypt isolation, which is described in the following section. The "SFT" samples were suspended in 0.5 mL of a freezing medium (10% dimethyl sulfoxide (DMSO) in 90% fetal bovine serum (FBS)), placed in 1 mL cryovials, and subjected to slow freezing at -80˚C overnight

using a freezing container (Thermo Fisher Scientific) [24,25]. The "FFT" samples were directly immersed in liquid nitrogen for 30 seconds in 1 mL cryovials [25]. Both "SFT" and "FFT" samples were then stored at -80˚C for approximately 6 months. Prior to crypt isolation, the cryopreserved tissues were rapidly thawed at 37˚C and underwent three sterile washes [15,26,27].

## Crypt isolation and 3D organoid generation

Intestinal crypts were isolated from all tissue samples and cultured to generate organoids following the technique described previously [4]. Briefly, the tissue samples were minced to small fragments in 20 mM EDTA solution using sharp-pointed dissecting scissors (Fisher Scientific). The fragmented tissue was collected into a 15 mL conical tube and incubated for 1 hour at 4˚C on a tube rotator (Fisher Scientific). The crypt-containing supernatant was collected in a new 15 mL conical tube and pelleted by centrifugation at 200 x $g$ at 4˚C for 5 minutes. The pellet containing the crypts was resuspended in Matrigel (Corning) and the crypts were then seeded onto a 24-well plate in 30 μL per well.

After Matrigel was polymerized at 37˚C for 10–15 minutes, each well received 500 μL of Advanced DMEM/F12 (Gibco)-based organoid culture medium supplemented with 10% (vol/vol) Noggin conditioned medium, 20% (vol/vol) R-spondin conditioned medium, 100 ng/mL recombinant murine Wnt-3a (PeproTech), 500 nM A-83-01 (Sigma-Aldrich), 1x B27 supplement (Gibco), 50 ng/mL murine Epidermal Growth Factor (EGF) (R&D Systems), 10 nM Gastrin (Sigma-Aldrich), 1x N2 MAX Media supplement (R&D Systems), 10 mM Nicotinamide (Sigma- Aldrich), 1 mM N-Acetyl-L-cysteine (MP Biomedicals), 10 μM SB202190 (Sigma-Aldrich), 100 μg/mL Primocin (InvivoGen), 1x Penicillin/Streptomycin (Gibco), 2 mM Gluta-MAX (Gibco), and 10 mM HEPES (Gibco) [4]. Noggin and R-Spondin conditioned media were obtained by cultivating HEK293 cells engineered to secrete Noggin (Baylor's College of Medicine) [28] and Cultrex HA-R-Spondin1-Fc 293T cells (R&D Systems) as described previously [29]. Additionally, 10 μM Y-27632 (StemCellTechnologies) and 100 nM CHIR99021 (Sigma-Aldrich) were supplemented to the culture medium during the first few days of culture until organoid growth was noted in Matrigel. The plate was incubated at 37˚C, 5% $CO_2$ and the culture medium was replaced every two days for growth and maintenance of organoids.

Growth of organoids was monitored daily and recorded using phase-contrast microscopy (DMi1, Leica). The surface area of developing organoids was measured using ImageJ 1.54d (National Institute of Health) [30]. Since the number and quality of the isolated crypts varied between the tissue processing techniques, i.e. fresh or cryopreservation, the techniques were considered feasible or "success" when at least one organoid developed following the crypt isolation, consistently expanded through serial passages, and stably maintained for more than five passages.

## 3D organoid subculture and maintenance

Organoids were serially passaged and expanded every 4–6 days following previously described protocol following initial culture [4]. Briefly, organoids were recovered from Matrigel by replacing the culture medium with cold Cell Recovery Solution (Corning), incubating them for 30 minutes at 4˚C, and centrifuging them at 200 x $g$ at 4˚C for 5 minutes. Pelleted organoids were resuspended into TrypLE Express (Gibco) and disrupted by incubating in 37˚C water bath for 1 minute. Basal medium (Advanced DMEM/F12 with Glutamax, HEPES, and Penicillin/Streptomycin) was added to stop enzymatic disruption and the dissociated organoids were collected by centrifuging at 200 x $g$ at 4˚C for 5 minutes. The pellet was resuspended in Matrigel at an expansion ratio of approximately 1 to 6–8 wells and cultured in 30 μL per well as described above in 48-well plates with 300 μL of culture medium.

## Development of organoid-derived 2D monolayer

Organoids were recovered from Matrigel as described above and disrupted with TrypLE Express by incubating in 37˚C water bath for 10 minutes. Enzymatic reaction was stopped by adding basal medium and the cells were filtered through a 70 μm cell strainer (Fisher Scientific) to obtain single cell suspension. The single cells were seeded at $3 \times 10^5$ cells/well on 2% (vol/vol) Matrigel coated 24-well cell culture insert with pore size of 0.4 μm (0.33 cm$^2$, Falcon). Cells were cultured in 200 μL (apical chamber) and 500 μL (basal chamber) of the organoid culture medium supplemented with 20% (vol/vol) FBS, 10 μM Y-27632, 100 nM CHIR99021, and 500 nM LY2157299 (Sigma-Aldrich). The culture medium was replaced every two days and monolayer formation was monitored daily and recorded using phase-contrast microscopy (DMi1, Leica).

## Evaluation of epithelial barrier integrity

The barrier function of the intestinal epithelial monolayer was evaluated by measuring transepithelial electrical resistance (TEER) and performing permeability assay. The TEER was monitored daily using an epithelial Volt-Ohm Meter (Millicell ERS-2, Millipore AG) as described previously [31]. The measured value (Ω) was normalized by subtracting the value obtained from blank wells and multiplying by the surface area (cm$^2$) of the culture insert. Permeability assay was performed on Days 1, 3 and 5 of culture using 0.5 mg/mL 4 kDa fluorescein isothiocyanate (FITC)-dextran (Sigma-Aldrich) as described previously [32]. The fluorescence intensity of the culture medium in the basal chamber, which corresponds with the amount of the FITC-dextran that passed through the cell monolayer over 120 minutes, was measured with SpectraMax i3x microplate reader (Molecular Devices) with excitation and emission wavelengths of 495 and 535 nm, respectively. The apparent permeability ($P_{app}$) (cm/s) was calculated by dividing the amount of molecules that passed through the cell layer over a fixed time period (μg/s) by the initial FITC-dextran concentration in the apical chamber (μg/mL) and the surface area (cm$^2$) of the culture insert. Both TEER measurements and permeability assay were performed with at least two technical replicates using at least two biological replicates.

## Immunofluorescence staining

Cellular and structural characteristics of 3D organoids and 2D monolayers were evaluated by immunocytochemistry following the protocol described previously [4]. Briefly, the cells were fixed and permeabilized with 4% paraformaldehyde (Thermo Scientific) and 0.3% Triton X-100 (Thermo Scientific) for 15 minutes at room temperature, respectively, followed by blocking with 2% bovine serum albumin (BSA) (Cytiva) in PBS for 60 minutes. The initial 4% paraformaldehyde treatment also allowed 3D organoids to be recovered from the Matrigel. The cells were incubated with primary antibodies against E-cadherin (1:200, BD Biosciences), EpCAM (1:200, Abcam) and SOX9 (1:250, Abcam) diluted in 2% BSA overnight at 4˚C. Sambucus Nigra Agglutinin (SNA) (1:100, Vector Laboratories), a sialic acid-specific lectin which binds to mucin, was used to detect goblet cells. Subsequently, the cells were washed with PBS and incubated with a secondary antibody (Alexa Fluor 555-conjugated Goat Anti-Rabbit IgG H&L, 1:1000, Abcam) for 1 hour at room temperature. Furthermore, the cells were stained for nuclei (4',6-diamidino-2-phenylindole dihydrochloride (DAPI), 1:1000, Thermo Scientific) and F-actin (Alexa Fluor 647-conjugated phalloidin, 1:400, Invitrogen) following the manufacturers' recommendation. For 3D organoids, EdU assay (Invitrogen) was also performed following the manufacturer's protocol to detect actively proliferating cells.

The samples were washed with PBS and mounted on either glass bottom dishes (Matsunami) or slide glasses using ProLong Gold Antifade reagent (Invitrogen). A white light point

scanning confocal microscope (SP8-X, Leica) was used to capture fluorescent images, which were subsequently processed using LAS X (Leica). Immunocytochemistry was performed in at least two technical replicates using three biological replicates for both 3D organoids and 2D monolayers. For 3D organoids, at least four independent fields of view were randomly selected and the percent positive cell rates for SOX9, SNA and EdU were determined by normalizing the number of positive cells by the total number of nuclei.

## Gene expression analysis of 3D organoids

RT-qPCR was performed to evaluate gene expression levels of 3D organoids derived from fresh and cryopreserved tissues and the organoid-derived 2D monolayers as described previously [4]. Marker genes evaluated included *LGR5* (stem cells), *CHGA* (enteroendocrine cells), *LYZC* (Paneth cells), *MUC2* (goblet cells), and *FABP2* (intestinal epithelial cells). Primers used in this study are summarized in S1 Table [13,33,34]. Briefly, total RNA was extracted from 3D organoids and 2D monolayers using RNeasy Plus Mini Kit (Qiagen) and cDNA was synthesized using a High-Capacity cDNA Reverse Transcription Kits (Applied Biosystems). RT-qPCR reactions was carried out using PowerUp SYBR Green Master Mix (Applied Biosystems), with 40 amplification cycles at 60˚C. The standard curve method was applied to calculate relative gene expression using *GAPDH*, *RPL0*, and *ACTB* as internal control [33,35–37]. The mean of these genes for each sample was taken to provide an optimum basis for a normalization of the target genes [38,39]. RT-qPCR reactions were carried out in three technical replicates using three biological replicates.

## Scanning and transmission electron microscopy

2D monolayers were prepared for scanning electron microscopy (SEM) and transmission electron microscopy (TEM) imaging following the protocol described previously, with slight modifications [5]. Briefly, the samples were fixed with 2.5% (vol/vol) glutaraldehyde (Ted Pella) in 0.1 M sodium cacodylate buffer (Ted Pella) overnight at 4˚C. For SEM, the samples were washed in 0.1 M cacodylate buffer and fixed with 1% osmium tetroxide (Electron Microscopy Sciences) in 0.1 M sodium cacodylate buffer for 30 minutes at room temperature, followed by serial dehydration in 30–100% ethanol and hexamethyldisilazane (HDMS) (SPI Supplies). Subsequently, the samples were mounted on stubs, coated with Platinum/Palladium using a high-resolution sputter coater (Cressington), and imaged using Quanta 200F SEM (FEI). For TEM, the samples were washed in 0.1 M cacodylate buffer and fixed with 1% osmium tetroxide and 1% ferrocyanide (Braun Knecht Heimann Company) in 0.1 M sodium cacodylate buffer overnight at 4˚C, followed by staining with 2% uranyl acetate (SPI Supplies) and serial dehydration in 30–100% ethanol and propylene oxide (Electron Microscopy Sciences). The samples were then infiltrated with spurrs resin (Ted pella), polymerized and sectioned to 80 nm thickness for imaging using Tecnai G2 20 Twin TEM (FEI).

## Statistical analyses

Quantitative data were analyzed using R v.3.4.1 (The R foundation) and plotted using GraphPad Prism 10.1.1 (GraphPad Software). The data were compared between organoids derived from fresh and slow-frozen tissues using either Wilcoxon's signed rank test or paired t-test for independent samples upon evaluating each dataset for the normality using Shapiro-Wilk test. Results were presented as the mean ± standard error of the mean (SEM), with $p$-values $\leq 0.05$ being considered statistically significant.

## Results

### Bovine rectal 3D organoids generated from fresh and slow-frozen tissues

Bovine rectal organoids were successfully generated from fresh and slow-frozen tissues but not from flash-frozen tissues following crypt isolation (Fig 1A). While crypts isolated from fresh tissues appeared intact and grew to form organoids, those from cryopreserved tissues resulted in release of single cells. Subsequently, single cells from slow-frozen tissue grew to form organoids, yet those from flash-frozen tissue did not survive to form organoids by 7 days of culture. As such, the mean surface area of developing organoids was significantly greater with fresh tissues compared with slow-frozen tissues during the initial cultivation (P0) ($p < 0.001$) (Fig 1B).

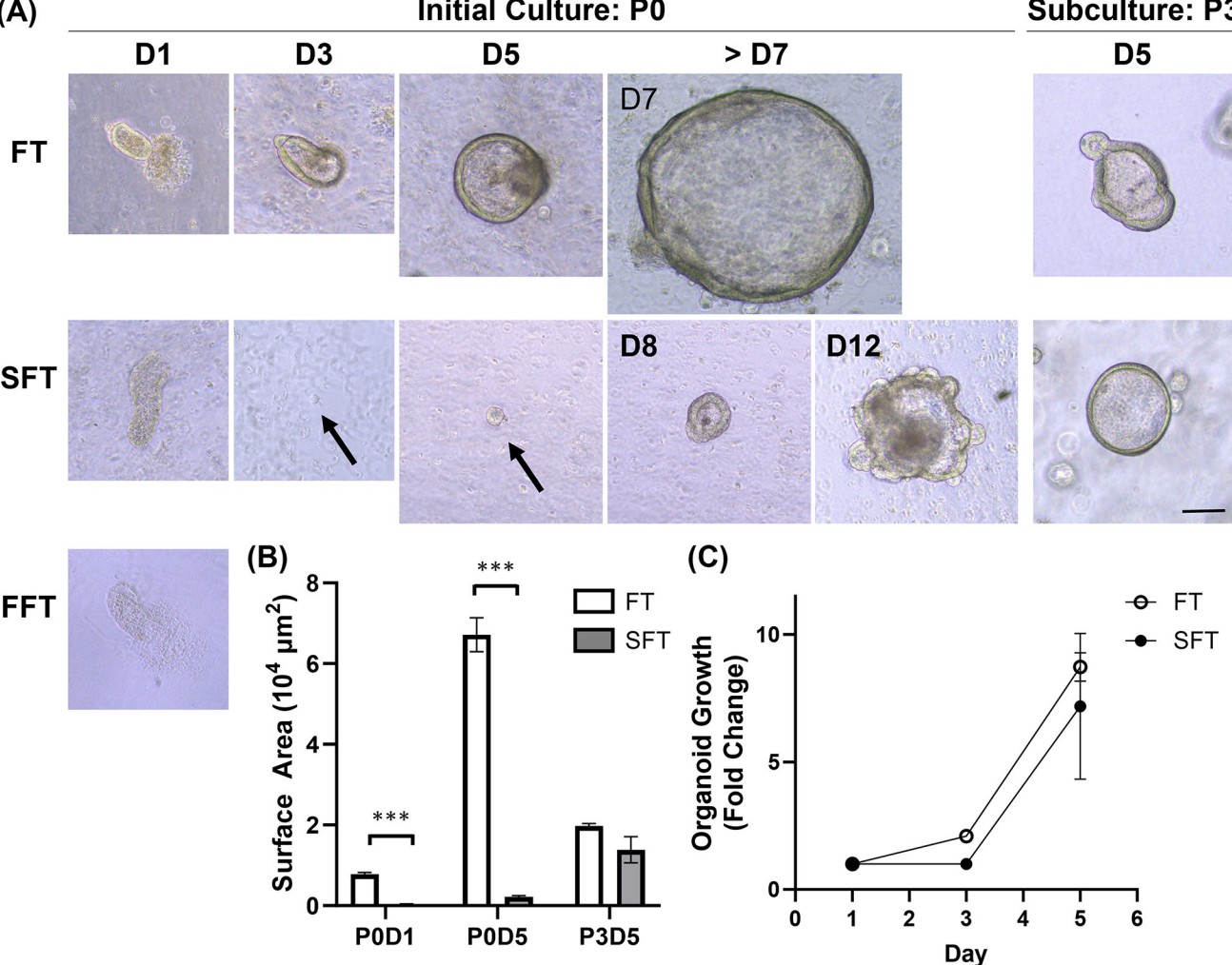

**Fig 1. Growth dynamics of bovine rectal organoids derived from fresh and cryopreserved tissues.** (A) Phase-contrast microscopy provided representative images showcasing isolated crypts 24 hours post-seeding in Matrigel (D1) and the evolution of organoids during the initial 1–2 weeks of culture in Matrigel (D3-12), across both the first (P0) and third passages (P3). Notably, organoids failed to develop from the flash-frozen tissue samples. (B&C) The surface area of growing organoids was measured on Days 1, 3, and 5 during P0, and on Day 5 during P3. The organoids derived from fresh tissues were significantly larger than those from slow-frozen tissues during P0 ($p < 0.001$), but this size discrepancy was not observed at P3 ($p = 0.24$) (B). Additionally, the growth rates of organoids during P0, expressed as fold changes relative to their size on Day 1, showed no significant differences between the groups (C). These measurements were based on 4 to 10 randomly selected organoids from a minimum of two biological replicates. The results are presented as mean ± standard error of the mean (SEM). Statistical analysis was performed with Wilcoxon's signed rank tests. FT, Fresh Tissue; SFT, Slow-Frozen Tissue; FFT, Flash-Frozen Tissue. Bar, 100 μm. *** $p < 0.001$.

However, actual growth rate did not differ from each other ($p$ = 0.25~0.5) (Fig 1C). The difference in the surface area became insignificant by passage 3 (P3) ($p$ = 0.24) and organoids from both groups appeared morphologically indistinguishable from each other under phase-contrast microscopy (Fig 1A and 1B). Organoids generated from both fresh and slow-frozen tissues continued to exhibit their proliferating capacity without showing any noticeable changes in their morphology or growth rates under *in vitro* culture condition using our organoid culture medium over 15 passages at the time of manuscript preparation.

## Structural and cellular characteristics of bovine rectal 3D organoids

Bovine rectal organoids derived from both fresh and slow-frozen tissues exhibited similar structural and cellular characteristics to each other based on immunofluorescence staining (Fig 2A). Organoids from both groups formed cystic luminal structure surrounded with polarized epithelium and epithelial lineage cells, which were characterized with basolateral adherence junction formation (E-cadherin), apical brush border (F-actin) and basal nuclei (DAPI). Positive EpCAM stain confirmed epithelial nature of the cells conforming the organoids. Furthermore, presence of mixed cell populations of epithelial cell lineages within organoids was demonstrated by identifying SOX9-positive stem cells, mucin producing SNA-positive goblet cells, and actively proliferating EdU-positive cells in both groups. The percentages of these cells within organoids relative to the number of nuclei were not different between the two groups ($p$ = 0.10~0.82) (Fig 2B).

## Gene expression of bovine rectal 3D organoids

Expression of epithelial lineage cell marker genes in bovine rectal organoids was determined with RT-qPCR and compared between the organoids derived from fresh and slow-frozen tissues (Fig 3). Significant difference was noted between the two groups in the expression levels of stem cell (*LGR5*, $p<0.05$), enteroendocrine cell (*CHGA*, $p<0.05$) and enterocyte (*FABP2*, $p<0.01$) marker genes. No difference was noted in Paneth cell (*LYZC*, $p$ = 0.65) and goblet cell (*MUC2*, $p$ = 0.82) marker genes.

## Development of stable bovine rectal organoid-derived 2D monolayer

To optimize the establishment of the 2D monolayer, the study experimented with different combinations of single cell densities derived from organoids and compositions of the maintenance culture media (S2 Table). The most effective and stable monolayer formation was achieved using a coating of 2% Matrigel, a seeding density of 3 x $10^5$ cells per well, and a medium supplemented with GSK3 inhibitor (CHIR99021) along with 20% FBS, ROCK inhibitor (Y-27632) and TGF-β receptor inhibitor (LY2175299) (S1 and S2 Figs). Stable bovine rectal 2D monolayer was successfully developed from organoids derived from both fresh and slow-frozen tissues and its structural and functional characteristics were evaluated (Fig 4). Cells seeded onto the cell culture inserts reached confluence by Day 2 of culture and formed monolayer, which was maintained up to Day 8 (Fig 4A). The integrity of the monolayer reached stable by Day 3 of culture and maintained up to Day 8 based on TEER measurement and permeability assay, with the mean TEER and $P_{app}$ values of 160.9±42.2 $\Omega^*cm^2$ and 1.1±0.7 x $10^{-7}$ cm/s, respectively, at Day 3 of culture (Fig 4B). Immunofluorescence staining of the monolayer for F-actin, E-cadherin and SNA demonstrated formation of apical brush border and basolateral adherence junctions as well as presence of mucus producing goblet cells (Fig 4C). RT-qPCR analysis at Day 6 of culture documented the expression of both stem and lineage cell marker genes in the cells that formed monolayer (Fig 4D).

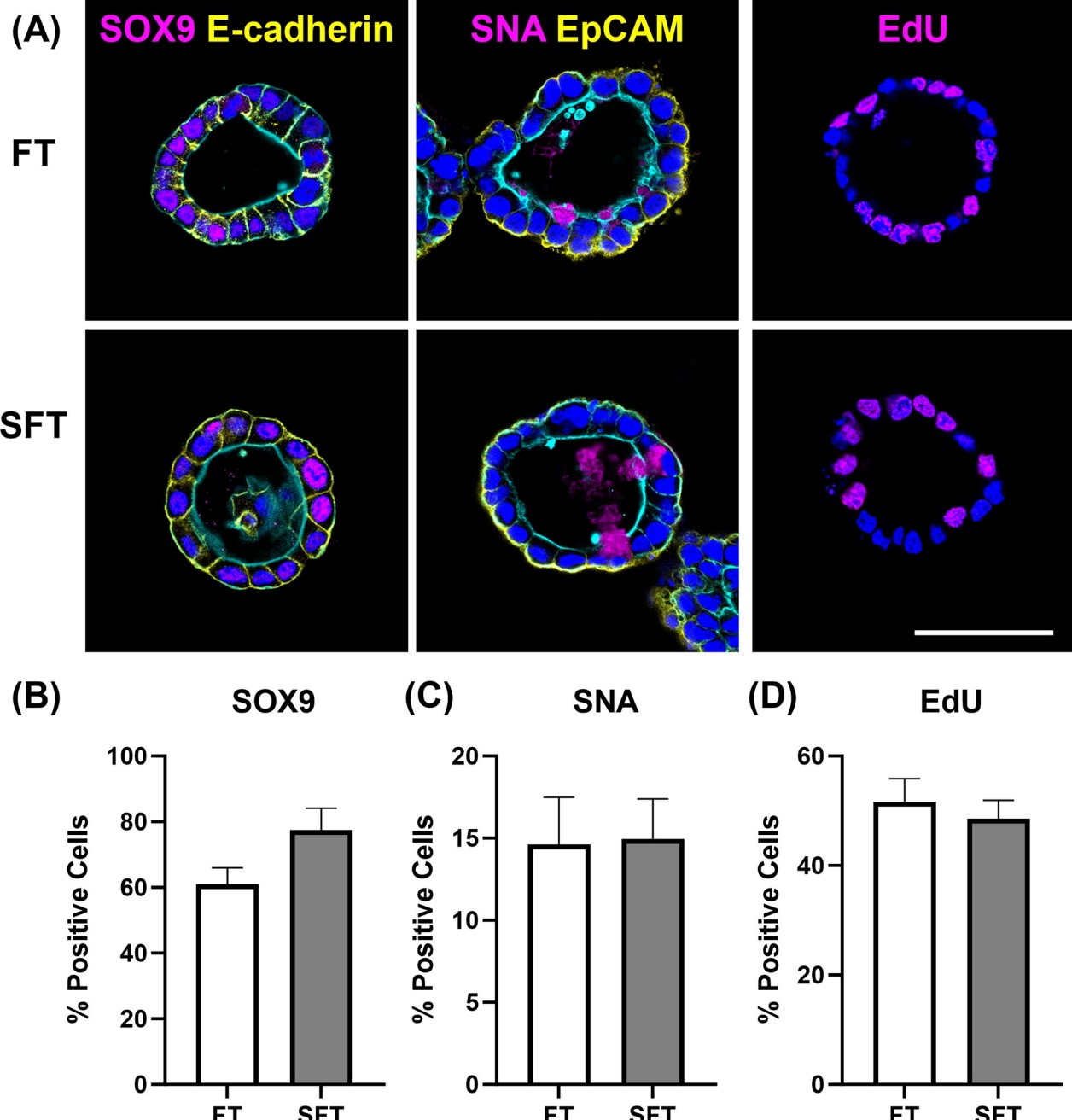

**Fig 2. Immunocytochemical characterization of fresh (FT) and slow-frozen tissue (SFT)-derived bovine rectal organoids.** (A) Structural and cellular characteristics of organoids were indifferent between the two groups. Confocal microscopy images demonstrated formation of cystic luminal structure with basolateral epithelial adherens junctions (E-cadherin, yellow), apical brush border (F-actin, cyan) and basal nuclei (DAPI, blue), and presence of multi-cellular populations including stem cells (SOX9, magenta), mucin-producing goblet cells (SNA, magenta), epithelial cells (EpCAM, yellow), and actively proliferating cells (EdU, magenta). (B) The percent positive cells for SOX9, SNA and EdU were evaluated by normalizing the number of positively stained cells by the total numbers of nuclei. No difference was noted between the two groups in all markers using Wilcoxson's signed rank test (SOX9) or paired t-tests (SNA and EdU). 4 to 10 randomly selected organoids from at least two biological replicates were assessed. The results are presented as mean ± standard error of the mean (SEM). Bar, 50 μm.

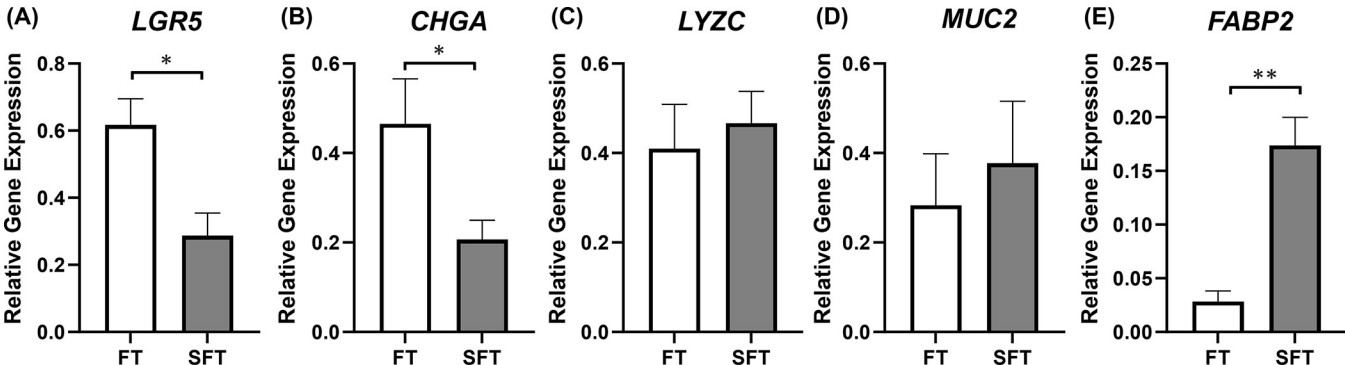

**Fig 3. Gene expression profiles of fresh (FT) and slow-frozen tissue (SFT)-derived bovine rectal organoids.** RT-qPCR was performed to determine expression of epithelial cell markers: Leucine rich repeat containing G protein-coupled receptor 5 (*LGR5*) for stem cells (A), Chromogranin A (*CHGA*) for enteroendocrine cells (B), Lysozyme C (*LYZC*) for Paneth cells (C), Mucin 2 (*MUC2*) for goblet cells (D), and fatty acid-binding protein 2 (*FABP2*) for enterocytes (E) in organoids at Day 5 of culture following three passaging. Three technical replicates from three biological replicates were used. The gene expression levels of each of the target genes were calculated relative to that of the internal control, which was the mean of *GAPDH*, *RPL0*, and *ACTB*. The results are presented as mean ± standard error of the mean (SEM). Statistical analysis was performed with either paired t-test (*LGR5* and *CHGA*) or Wilcoxon's signed rank test (*LYZC*, *MUC2*, and *FABP2*) for independent samples. * $p < 0.05$, ** $p < 0.01$.

SEM imaging revealed well developed microvilli on the apical surface of the monolayer (Fig 5A and 5B). Presence of goblet cells within the monolayer was also confirmed with SEM and TEM by visualizing characteristic goblet cell orifices and mucin granules as described

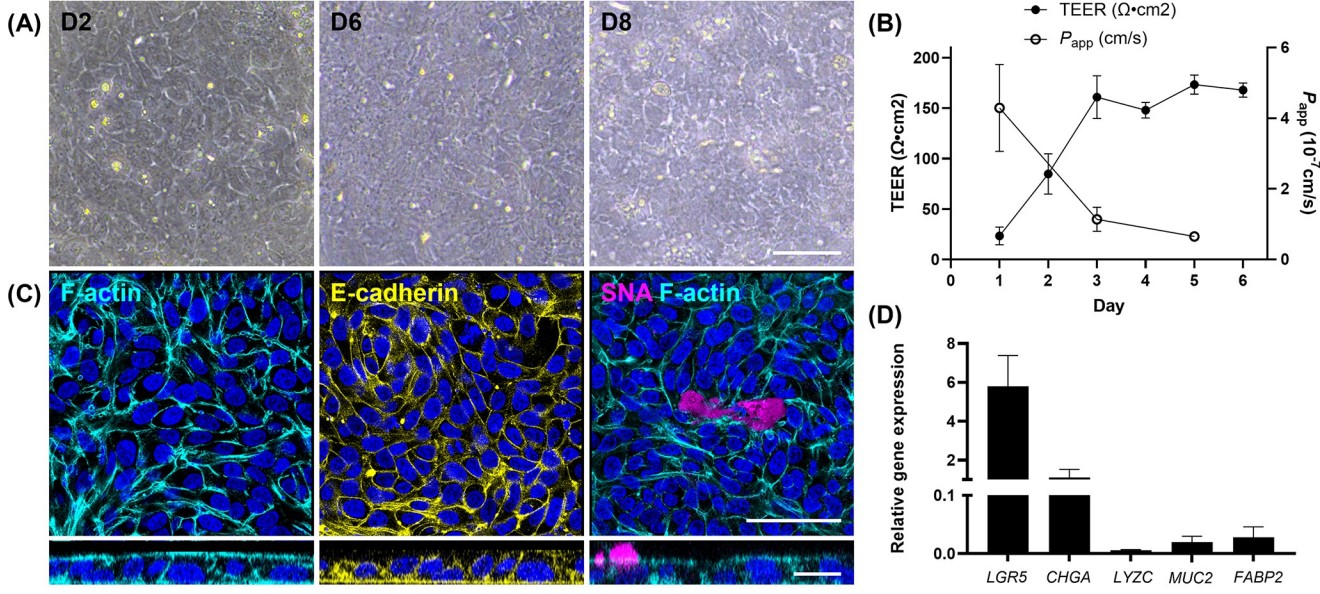

**Fig 4. Development of stable bovine rectal organoid-derived 2D monolayer.** (A) Representative phase-contrast microscopy images of 2D monolayer at Days 2, 6 and 8 (D2-8) of culture on a cell culture insert. (B) Transepithelial electrical resistance (TEER) and permeability assay using 4 kDa FITC-dextran demonstrated development of stable and functional epithelial barrier integrity by Day 3 of culture. The results are presented as mean ± standard error of the mean (SEM) from a minimum of two biological replicates with at least two technical replicates. (C) Immunofluorescent staining revealed uniform expression of F-actin (Phalloidin, cyan) and basolateral adherens junctions (E-cadherin, yellow), and presence of mucin-producing goblet cells (SNA, magenta). Nuclei are visualized by DAPI staining (blue). Top-down view and z-stack images are shown. Bars, 50 μm. (D) RT-qPCR of the monolayer at Day 6 of culture documented expression of stem and lineage cell marker genes. Two technical replicates from three biological replicates were evaluated. The gene expression levels of each of the target genes were calculated relative to that of the internal control, which was the mean of *GAPDH*, *RPL0*, and A*CTB*. *LGR5*: Leucine rich repeat containing G protein-coupled receptor 5, *CHGA*: Chromogranin A, *LYZC*; Lysozyme C, *MUC2*: Mucin 2, *FABP2*: Fatty acid-binding protein 2.

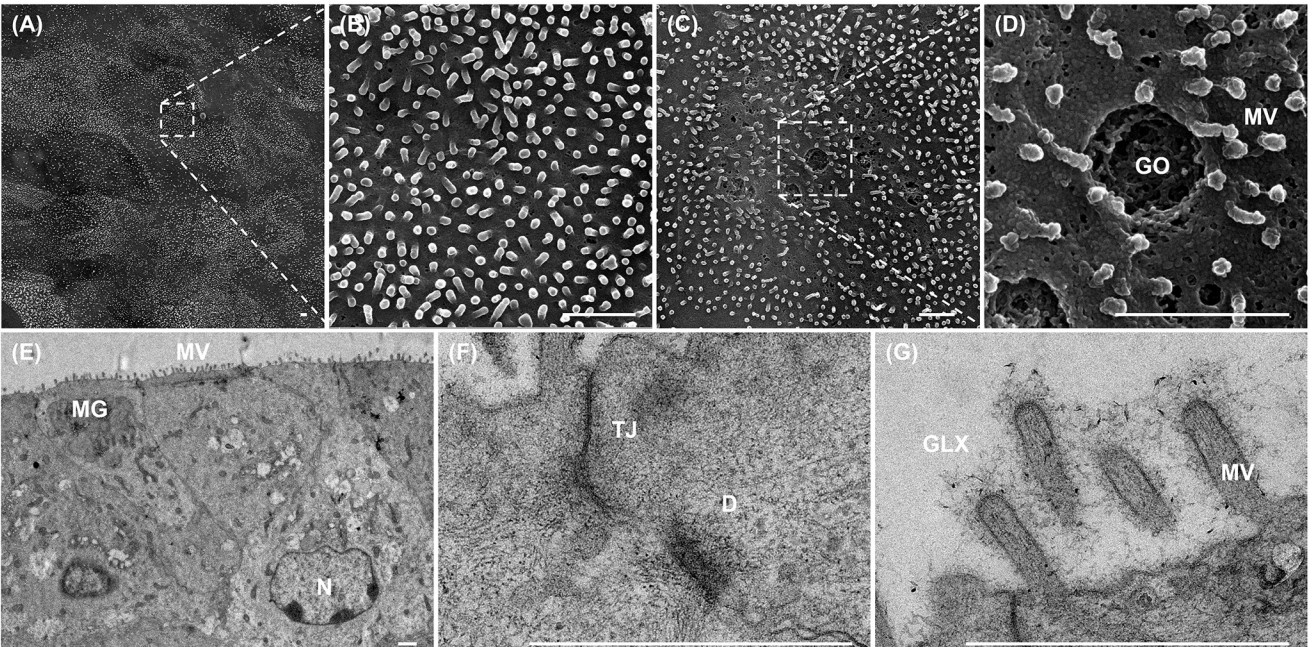

**Fig 5. Electron microscopic characterization of the bovine rectal organoid-derived 2D monolayer.** Representative scanning (A-D) and transmission (E-G) electron microscopy images of 2D monolayer on Day 5 of culture on a cell culture insert. The formation of microvilli on the apical surface of the monolayer was confirmed at low (A) and high (B) power magnifications. A goblet cell was identified on the apical surface of the monolayer at low power magnification (C), where a goblet cell orifice (GO) and microvilli were highlighted at high power magnification (D). (E) A low power magnification image revealed uniformly developed microvilli (MV) on the apical surface and a goblet cell containing multiple mucin granules (MG) in the cytoplasm. N = nucleus. (F&G) A high power magnification images highlight the formation of inter-cellular tight junctions (TJ), desmosome (D) and microvilli (MV) covered with glycocalyx (GLX). Bars, 1 μm.

previously in human and canine colonic monolayers (Fig 5C–5E) [5,40]. Furthermore, TEM imaging demonstrated formation of inter-cellular tight junctions and desmosomes as well as an apical brush border bearing the glycocalyx (Fig 5F and 5G) [5,41].

## Discussion

This study is the first to successfully demonstrate that bovine rectal organoids can be effectively generated from cryopreserved biopsy tissues using the slow freezing technique. The organoids generated from slow-frozen tissues in this study were successfully maintained and expanded for long term in *in vitro* environment using the previously described Advanced DMEM/F12 based culture medium [4]. Furthermore, employing 48-well plates for all subsequent subcultures, in conjunction with the effective use of DMEM/F12 based culture medium, significantly enhanced the efficiency and cost-effectiveness of the organoid expansion process. This approach reduced the culture medium requirement by 200 μL per well, offering a notable advantage over the use of 24-well plates.

While initial sizes of the organoids derived from slow-frozen tissues were much smaller compared to those derived from fresh tissues, this discrepancy diminished in subsequent passages, as noted previously [15,26]. This observation indicates that tissue cryopreservation would not pose significant impact on subsequent research using organoids. The structural and morphological features of the organoids from cryopreserved tissues closely resembled those from fresh tissues of the same donors, as confirmed through phase-contrast and confocal microscopy. This observation is consistent with previous studies involving human intestinal tissues and various other organs [15,16].

RT-qPCR analysis of the 3D organoids revealed some differences in the gene expression profiles of stem and epithelial lineage cell markers between the two groups. The observation of upregulated lineage cell markers and downregulated stem cell markers in organoids derived from slow-frozen tissue suggests a degree of differentiation. This finding aligns with similar alterations observed in cryopreserved enteric neurospheres when compared to their fresh counterparts [42]. Although the specific causes of these alterations require further investigation, the changes noted in this study may be partially attributed to the differentiation-promoting effects of DMSO present in the freezing medium [43,44]. Nonetheless, the organoids from slow-frozen tissues retained a heterogeneous cell population similar to their fresh-tissue counterparts. These findings underscore the effectiveness of the slow freezing technique as a method for preserving bovine rectal tissues for organoid development. This approach offers geographical flexibility and year-round accessibility to viable tissue samples. It also provides a valuable *in vitro* tool for biomedical, agricultural, and veterinary research, with the ethical advantage of reducing the number of animals needed for studies.

The study demonstrates a remarkable 100% success rate in generating organoids from slow-frozen biopsy tissues, showcasing the effectiveness of this cryopreservation method. This outcome not only matches but may also surpass previous results achieved with cryopreserved human intestinal tissues, which reported success rates ranging from approximately 72% to 100% in organoid development [15,45]. In contrast, the current study observed no success in developing organoids from flash-frozen tissue samples. This finding is notably different from earlier studies that reported successful organoid generation using flash-frozen mouse intestinal tissues and human breast cancer samples [25,46]. Slow freezing is a commonly used cryopreservation technique where tissue samples are immersed in a freezing medium consisting of 5–10% DMSO in culture medium and/or FBS and the temperature is gradually reduced down to -80˚C [15,25,45]. On the other hand, the flash freezing method, another popular approach for tissue biobanking and validated for organoid generation, has shown inconsistent use of freezing medium across different studies [25,46]. Notably, in this study, no freezing medium was employed for flash freezing the tissue samples. The absence of a freezing medium in flash freezing, unlike in the slow freezing process, might be a contributing factor to the reduced tissue viability during freeze-thaw cycles, potentially affecting the success rate of organoid generation. Incorporating a freezing medium in the flash freezing process, similar to the slow freezing technique, might enhance tissue preservation and improve the likelihood of successful organoid development [46].

Another notable accomplishment of this research is the successful establishment of a functional 2D monolayer derived from bovine rectal organoids. The integrity and stability of the epithelial barrier in the monolayer were confirmed through the documentation of stable TEER and $P_{app}$ values after three days of culture. The observed TEER value of approximately 161 $\Omega$*$cm^2$, while slightly lower than those reported in previous studies with bovine ileal and colonic organoid-derived monolayers, aligns with the presence of diverse cell types like mucus-producing goblet cells in the culture [12,13,31]. Such cellular diversity is known to influence TEER values, potentially rendering the model more physiologically relevant [31]. Interestingly, the TEER values in this study are comparable to those observed in co-culture models of Caco-2 and mucus-producing TH29-MTX cells [47]. It has been described that TEER measurements have been shown to be affected by a number of factors including temperature, cell types, cell passage number and culture medium composition [31]. Thus, taking these factors into consideration is important when comparing the results reported from different laboratories. In summary, the development of a functional epithelial barrier in *in vitro* intestinal monolayer models should not solely focus on achieving high TEER values or matching values reported by other laboratories. Instead, a more accurate indicator of barrier

formation may be the observation of a decreasing trend in $P_{app}$ values reaching a plateau, alongside increasing TEER values until they stabilize, as demonstrated in this and other studies [5].

The imaging of the 2D rectal organoid-derived monolayer, achieved through a range of microscopy techniques including SEM and TEM, revealed significant findings. It confirmed the presence of mucus-producing goblet cells and the development of a physiological brush border interface, characterized by numerous microvilli on the polarized epithelium. This interface is further interconnected with tight junctions and desmosomes, mirroring characteristics typically seen in intestinal epithelium. These observations are consistent with previous studies in canine and human intestinal monolayers, supporting the model's accurate representation of *in vivo* intestinal structures [5,40,48–50].

The detection of epithelial lineage cell marker gene expressions via RT-qPCR corroborates their physiological similarity to the native intestine. The patterns of relative expression levels for markers of enterocytes, goblet cells, and Paneth cells within the monolayers closely mirror the distribution of these differentiated cell types as observed in the human lower intestine [51]. Moreover, the formation of a physiologically functional epithelial barrier in this model underscores its potential as an effective *in vitro* tool. Particularly, the bovine rectal organoid-derived 2D monolayer offers novel opportunities for research that benefits from enhanced access to the apical surface. This includes studies in host-pathogen interactions, host innate immunity, and the functions of drug and nutrient transport, thereby broadening the scope of its applicability in various research domains.

Intriguingly, this study discovered that the 2D monolayer derived from bovine rectal organoids forms and sustains more effectively under specific conditions: a higher cell seeding density and increased FBS concentrations in the culture medium compared to that used in a previous study with bovine ileal organoid-derived monolayers ($2.5 \times 10^4$ cells/0.33 cm$^2$/well, 1% FBS) [13]. A similar adaptation was observed in another study on porcine ileal organoids, which achieved successful monolayer formation at a lower seeding density ($2.5 \times 10^4$ cells/0.33 cm$^2$/well) but required a comparably high FBS concentration of 20%, as in the current study [41]. In the current model, the addition of 20% FBS, along with the ROCK, TGF-β receptor, and GSK3 inhibitors to the organoid culture medium, was instrumental in enhancing cell adhesion to the cell culture insert and maintaining an intact monolayer for an extended period. Conversely, reducing or completely removing FBS and GSK3 inhibitor led to premature detachment of the monolayer. Similarly, lower seeding densities, akin to those in previous studies, did not result in uniform monolayer formation within four days of culture. The exact reasons for these variations are not entirely clear, given the numerous differences between these studies, such as organoid processing techniques and other aspects of culture medium composition beyond FBS supplementation. However, these findings suggest that optimal culture conditions might vary based on the intestinal segment and species from which the cells are derived, as has been observed in 3D organoid cultures [4,52,53]. Likewise, optimum culture conditions which actively induce cellular differentiations while effectively maintaining the epithelial monolayers would be worth investigating in the future studies as it could provide an even superior *in vitro* model to study biological and physiological functions of the intestinal barrier as evaluated in 3D organoids [4].

## Conclusions

This study successfully demonstrated the viability of generating bovine rectal organoids from cryopreserved biopsy tissue, alongside establishing an optimized and robust method for developing a functional rectal monolayer derived from adult bovine rectal organoids. Given that the

bovine rectum is a key site targeted by numerous enteric pathogens of significant public health concern, such as EHEC and *Salmonella spp.*, the absence of a physiologically relevant *in vitro* bovine rectal monolayer model until now has been a notable gap. The validated techniques in this research provide a potent tool for advancing studies using bovine rectal organoids and their derivative 2D monolayers. The ability to cryopreserve viable tissues presents a valuable opportunity for establishing a tissue biobank that can be accessed as needed, thereby reducing the reliance on live animals and associated costs for multiple studies. The methodologies introduced in this study are poised to enhance basic research into normal intestinal physiology and support applied and translational research endeavors, such as exploring disease development mechanisms, host-pathogen interactions, and host immune responses. Furthermore, applying these techniques to other intestinal segments could significantly enrich our understanding of bovine intestinal physiology, contributing profoundly to bridging the existing knowledge gap in bovine organoid technology.

## Supporting information

**S1 Fig. Optimization of bovine rectal organoid-derived 2D monolayer culture condition.** Representative phase-contrast microscopy images of 2D monolayer on Days 2, 4 and 6 (D2-6) of culture. Stable monolayer formation was achieved consistently when the cells were seeded at a density of 3–5 x $10^5$ cells per 24-well culture insert and cultured in the medium supplemented with GSK3 inhibitor (+CHIR99021). Asterisks denote areas where monolayer was not achieved or disrupted following initial monolayer formation. Bar, 100 μm.
(TIF)

**S2 Fig. Epithelial barrier integrity of bovine rectal organoid-derived 2D monolayers.** The most stable transepithelial electrical resistance (TEER) was observed when the cells were seeded at a density of 3 x$10^5$ cells per 24-well culture insert and cultured in the medium supplemented with GSK3 inhibitor (+CHIR99021). The results are presented as mean ± standard error of the mean (SEM) from two technical replicates for each condition.
(TIF)

**S1 Table. Primers used to evaluate gene expression of bovine rectal organoids.** Gene name, forward and reverse sequences and references are listed.
(TIF)

**S2 Table. Culture conditions evaluated during optimization of bovine rectal organoid-derived 2D monolayer culture.** Conditions of extracellular matrix (ECM) coating, cell seeding density and culture medium compositions were listed together with the maximum transepithelial electrical resistance (TEER) value that was achieved under each culture condition.
(TIF)

## Acknowledgments

The authors would like to thank the participating slaughterhouse for supplying donor cattle. The authors also would like to thank Itsuma Nagao and Nao Nagao-Akiyama for their help in tissue sampling, and Dr. Valerie Lynch-Holm and Dr. Brittney Wager from Franceschi Microscopy and Imaging Center at Washington State University for their technical support in confocal and electron microscopy.

## Author Contributions

**Conceptualization:** Minae Kawasaki, Yoko M. Ambrosini.

**Data curation:** Minae Kawasaki.

**Formal analysis:** Minae Kawasaki, Yoko M. Ambrosini.

**Funding acquisition:** Yoko M. Ambrosini.

**Investigation:** Minae Kawasaki.

**Methodology:** Minae Kawasaki, Yoko M. Ambrosini.

**Project administration:** Yoko M. Ambrosini.

**Resources:** Yoko M. Ambrosini.

**Software:** Yoko M. Ambrosini.

**Supervision:** Yoko M. Ambrosini.

**Validation:** Minae Kawasaki.

**Visualization:** Minae Kawasaki.

**Writing – original draft:** Minae Kawasaki.

**Writing – review & editing:** Minae Kawasaki, Yoko M. Ambrosini.

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
