## [Decision Letter · Decision Letter 0]

19 Feb 2024

PONE-D-23-43133Accessible luminal interface of bovine rectal organoids generated from cryopreserved biopsy tissuesPLOS ONE

Dear Dr. Ambrosini,

Thank you for submitting your manuscript to PLOS ONE. After careful consideration, we feel that it has merit but does not fully meet PLOS ONE’s publication criteria as it currently stands. Therefore, we invite you to submit a revised version of the manuscript that addresses the points raised during the review process.

We look forward to receiving your revised manuscript.

Kind regards,

Yash Gupta, Ph.D.

Academic Editor

PLOS ONE

Journal Requirements:

2. We note that Figure 1 in your submission contain copyrighted images. All PLOS content is published under the Creative Commons Attribution License (CC BY 4.0), which means that the manuscript, images, and Supporting Information files will be freely available online, and any third party is permitted to access, download, copy, distribute, and use these materials in any way, even commercially, with proper attribution. For more information, see our copyright guidelines: http://journals.plos.org/plosone/s/licenses-and-copyright.

Additional Editor Comments:

Authors need to incorporate reviewers' suggestions or dispute it with an apt rebuttal. The manuscript is well written and I concur with the reviewers that a minor revision is required

Reviewers' comments:

Reviewer's Responses to Questions

**Comments to the Author**

1. Is the manuscript technically sound, and do the data support the conclusions?

Reviewer #1: Yes

Reviewer #2: Yes

2. Has the statistical analysis been performed appropriately and rigorously? 

Reviewer #1: Yes

Reviewer #2: No

3. Have the authors made all data underlying the findings in their manuscript fully available?

Reviewer #1: Yes

Reviewer #2: Yes

4. Is the manuscript presented in an intelligible fashion and written in standard English?

Reviewer #1: Yes

Reviewer #2: Yes

5. Review Comments to the Author

Reviewer #1: In this manuscript, the authors detail methods for creating bovine rectal organoids from both freshly dissected tissues and 'slow-frozen' cryopreserved samples. The manuscript is well-prepared, with data presented meticulously, including proper statistical analysis and satisfactory technical and biological replicates.

The authors should clarify the rationale behind using three different genes (GAPDH, RPL0, and ACTB) as internal controls. Additionally, specify which gene was used as the internal control for Figure 4.

The authors observed no significant differences in the behaviors (growth rates) and characteristics (morphology, etc.) of organoids obtained from freshly dissected tissues and 'slow-frozen' cryopreserved tissue samples after three generations (P3). However, they need to address why the mRNA level of LGR5 (a stem cell marker) is significantly down-regulated, while FABP2, an enterocyte marker, is significantly up-regulated in organoids generated from 'slow-frozen' cryopreserved tissues.

In Fig.5E, the SNA staining signals (in green color) appear outside the cells, suggesting potential artifacts. Additionally, in line with the RT-qPCR results, the authors may consider using MUC2 immunofluorescence staining for goblet cells.

The manuscript lacks evaluation of organoids in the differentiated status, which is crucial for understanding the biological and physiological functions of the intestinal epithelial barrier. The authors should address this point in their discussion.

Reviewer #2: The manuscript submitted by Kawasaki & Ambrosini represents an important piece of work in the field of organoid research. Their work on the characterization of bovine rectal organoids holds particular relevance for further studies related to animal welfare and addressing intestinal infections that affect these animals. The manuscript has been diligently composed, showcasing a well-defined study design and the utilization of suitable research methodologies. However, in order for it to align with the requisite standards for publication, we respectfully recommend the incorporation of the following suggestions to enhance the quality and reproducibility of the research.

Line 78 ongoing: When mentioning the collection of the tissues the authors may specify the localization of the samples, which were harvested. Were these samples harvested all at the same location or spread over a wider area of the bovine rectum?

Line 92: Is the medium used for the ´sterile wash´ the same washing medium as described above (PBS including Penicillin/streptomycin and Gentamicin)?

Line 95: The authors state the crypt isolation as “…described previously [4]”. Please add another phrase stating, that the isolation is described in the following.

Line 108: There is a typo “pelted”

Line 109: Are the crypts seeded after centrifugation regardless their number? It is unlikely to expect, the all harvested material contained the same amount of crypts. Since the development form crypts to organoids and the later cultivation of the organoids is highly dependent of the number of crypts/organoids per Matrigel droplet, it is mandatory to compare cultivation methods with the same amount of crypts.

Line 112: The medium is called DMEM/F12 not DMED/F12

Lines 134-136: How large were the Matrigel droplets, which were placed, in the 48-well plates? The same size as in the 24-well plates mentioned earlier? Why the switch in plate format?

Lines 154-157: One does not measure the number of molecules but the emission as a measure for the amount of transport fluorescent dye.

Line 166: What was the solvent of BSA?

Line 175: Please state how the organoids were dissolved from the Matrigel in order to mount them on glass bottom dishes.

Line 185: Please change the writing to “marker genes”

Lines 209-214: Applying a Wilcoxon test requires testing for normal distribution of the data and used when this is not the case. Please include the statement why this test is used.

Figure 5E: The quality of the images is poor. Please change to images with better resolution to further improve the quality of the manuscript.

Lines 343-346: Please remove one of the redundant sentences.

Line 349: What does a 100% success rate mean? Just the generation of an indifferent number of organoids from crypts (quantity)? Or the successful generation from organoids from crypts regardless their numbers (quality).

Lines 396-398: If doing comparisons to other studies, please use more than one reference. Moreover, the amount of cells/cm2 is relevant, not the total number of cells.

6. PLOS authors have the option to publish the peer review history of their article (what does this mean?). If published, this will include your full peer review and any attached files.

Reviewer #1: **Yes: **Wei Ding

Reviewer #2: **Yes: **Gemma Mazzuoli-Weber

---

## [Author Response · Author response to Decision Letter 0]

7 Mar 2024

7 March 2024

Author response to review queries: PONE-D-23-43133

Dear Dr. Yash Gupta,

Thank you for your decision letter containing comments, queries and suggestions to our manuscript entitled “Accessible luminal interface of bovine rectal organoids generated from cryopreserved biopsy tissues" submitted to PLOS One. We have carefully considered each comment and have systematically addressed the queries and suggestions provided. Modifications have been made to the manuscript as necessary to reflect these recommendations. Below, we present our detailed responses to the feedback received. 

Kind regards,

Minae Kawasaki (on behalf of the authors)

Response:

Editors:

1. We note that Figure 1 in your submission contain copyrighted images. All PLOS content is published under the Creative Commons Attribution License (CC BY 4.0), which means that the manuscript, images, and Supporting Information files will be freely available online, and any third party is permitted to access, download, copy, distribute, and use these materials in any way, even commercially, with proper attribution. We require you to either (1) present written permission from the copyright holder to publish these figures specifically under the CC BY 4.0 license, or (2) remove the figures from your submission:

To address the raised concerns regarding copyrighted image use, we have made the decision to remove Figure 1 from the revised version of our manuscript. Unfortunately, we were able to obtain only a CC-BY-NC-ND license from BioRender.com, which does not meet the CC-BY 4.0 license requirements. We have modified the labelling of each figure from Fig 1-5 to Fig 1-4 where relevant (Revised manuscript lines 245, 270, 291, 320).

Reviewer #1: 

1. The authors should clarify the rationale behind using three different genes (GAPDH, RPL0, and ACTB) as internal controls. Additionally, specify which gene was used as the internal control for Figure 4.

In our study, the selection of housekeeping genes for normalization purposes was guided by the principle of achieving a stable expression level across all samples. This approach aligns with the methodologies proposed by Inderwies, et al. (2003) and Ontsouka, et al. (2004), who advocate for the utilization of multiple reference genes to enhance normalization accuracy. Consequently, three specific genes were identified based on their documented constant expression in previous research on the bovine intestine, as evidenced by studies from Shakya, et al. (2023), Charavaryamath, et al. (2011), Koch, et al. (2019), and Coelho, et al. (2022). The average expression levels of these three genes were then employed as a basis for normalizing the expression levels of the target genes under investigation. To clarify these points, we have added a sentence to the Materials and Methods section and modified the figure legend as below. We appreciate the opportunity to clarify these details and address any potential misinterpretations concerning our experimental approaches.

- Revised manuscript lines 200-202: “…to calculate relative gene expression, using GAPDH, RPL0, and ACTB as internal control (Shakya, et al, 2023; Charavaryamath, et al., 2011; Koch, et al., 2019; Coelho, et al., 2022). The mean of these genes for each sample was taken to provide an optimum basis for a normalization of the target genes (Ontsouka, et al., 2004; Inderwies, et al., 2003).”

- Revised manuscript line 297 (Fig 3 legend) : “The gene expression levels of each of the target genes were calculated relative to that of the internal control, which was the mean of GAPDH, RPL0, and ACTB.”

2. The authors observed no significant differences in the behaviors (growth rates) and characteristics (morphology, etc.) of organoids obtained from freshly dissected tissues and 'slow-frozen' cryopreserved tissue samples after three generations (P3). However, they need to address why the mRNA level of LGR5 (a stem cell marker) is significantly down-regulated, while FABP2, an enterocyte marker, is significantly up-regulated in organoids generated from 'slow-frozen' cryopreserved tissues.

We are grateful for the opportunity to address the oversight and enrich the discussion of our results. Previous research has indicated that DMSO can influence cellular differentiation (Adler, et al., 2006; Ogaki, et al., 2015). The variability in observed effects across studies, including ours, may not be straightforward to explain. However, we hypothesize that the presence of DMSO in our freezing medium might have contributed to these observations. In response to the reviewer's insightful query, we have revised the Discussion section accordingly:

- Revised manuscript lines 369-374: “The observation of upregulated lineage cell markers and downregulated stem cell markers in organoids derived from slow-frozen tissue suggests a degree of differentiation. This finding aligns with similar alterations observed in cryopreserved enteric neurospheres when compared to their fresh counterparts (Heumuller-Klug, et al., 2023). Although the specific causes of these alterations require further investigation, the changes noted in this study may be partially attributed to the differentiation-promoting effects of DMSO present in the freezing medium (Adler, et al., 2006; Ogaki, et al., 2015).”

3. In Fig.5E, the SNA staining signals (in green color) appear outside the cells, suggesting potential artifacts. Additionally, in line with the RT-qPCR results, the authors may consider using MUC2 immunofluorescence staining for goblet cells.

SNA, a lectin specific to sialic acid, binds to mucin secreted by goblet cells and is utilized for detecting intestinal mucus. Its correlation with MUC2 staining patterns has previously been documented in mouse intestinal tissues and human cell lines (Owen, et al., 2017). The extracellular manifestation of SNA staining signals aligns with expectations for mucus secretion from goblet cells, consistent with findings reported in earlier studies (refer to Fig 3 in Nelli, et al., 2010 and Punyadarsaniya, et al., 2011). Consequently, we argue that the positive signals observed in our research - specifically, the localization of signals to distinct cells across the monolayer and on the apical surface - represent genuine positives rather than artifacts.

The selection of antibodies suitable for identifying differentiated cell markers, such as goblet cell markers, is notably scarce for bovine species. While MUC2 staining in bovine intestinal organoids has indeed been reported (refer to Fig 3 in Lee, et al., 2021 and Park, et al., 2022), it is important to emphasize that the application of MUC2 in bovine models remains limited. Additionally, upon review, the image quality and staining specificity in these previous publications do not meet the high standards we aim for. In contrast, our findings demonstrate that SNA staining offers markedly enhanced specificity and image clarity in comparison to MUC2, as detailed in our recent publication (Kawasaki, et al., 2023). This distinction underscores the superiority of SNA for our purposes, highlighting its advantages in both specificity and visual quality. We have included explanation of SNA to enhance reader’s understanding in the Materials and Methods section. We have also made adjustments to the image by altering the color of each marker for consistency with the color scheme used in Fig 2.

- Revised manuscript lines 176-177: “Sambucus Nigra Agglutinin (SNA) (1:100, Vector Laboratories), a sialic acid-specific lectin which binds to mucin, was used to detect goblet cells.”

References for SNA images:

- Nelli, et al., 2010: https://bmcvetres.biomedcentral.com/articles/10.1186/1746-6148-6-4

- Punyadarsaniva, et al., 2011: https://journals.plos.org/plosone/article?id=10.1371/journal.pone.0028429

References for MUC2 images:

- Lee, et al., 2021: https://www.mdpi.com/2076-2615/11/7/2115

- Park, et al., 2022: https://www.ejast.org/archive/view_article?pid=jast-64-6-1105

4. The manuscript lacks evaluation of organoids in the differentiated status, which is crucial for understanding the biological and physiological functions of the intestinal epithelial barrier. The authors should address this point in their discussion.

Our investigation did not intentionally pursue cellular differentiation by altering culture conditions, such as using a Wnt-depleted medium, which is a common approach in such studies (Sato, et al., 2011; Liu, et al., 2020). Nonetheless, we have effectively documented the existence of differentiated cell types, notably mucus-secreting goblet cells featuring well-structured microvilli and tight junctions, within the bovine rectal monolayers. This was achieved through detailed immunofluorescence staining and electron microscopy analyses. To further substantiate our observations, we performed RT-qPCR analysis to evaluate the differentiation status of the monolayers more thoroughly. This analysis confirmed the presence of both stem cell and lineage-specific marker genes, indicative of goblet cells, enteroendocrine cells, and absorptive enterocytes, by Day 6 of culture. These combined results affirm the differentiation extent of the bovine rectal organoid-derived monolayers in our optimized culture conditions. We value this opportunity to clarify further and showcase the differentiated nature of our bovine rectal organoid-derived monolayers. We have updated the Materials and Methods, Results and Discussion sections to incorporate this additional evaluation. The new result was presented in Figure 4 in the revised manuscript. 

- Revised manuscript line 194: “RT-qPCR was performed to evaluate gene expression levels of 3D organoids derived from fresh and cryopreserved tissues and the organoid-derived 2D monolayers as described previously [4].”

- Revised manuscript lines 316-318: “RT-qPCR analysis at Day 6 of culture documented the expression of both stem and lineage cell marker genes in the cells that formed monolayer (Fig 4D).”

- Revised manuscript lines 329-333 (Fig 4 legend): “(D) RT-qPCR of the monolayer at Day 6 of culture documented expression of stem and lineage cell marker genes. Two technical replicates from three biological replicates were evaluated. The gene expression levels of each of the target genes were calculated relative to that of the internal control, which was the mean of GAPDH, RPL0, and ACTB. LGR5: Leucine rich repeat containing G protein-coupled receptor 5, CHGA: Chromogranin A, LYZC; Lysozyme C, MUC2: Mucin 2, FABP2: fatty acid-binding protein 2.”

- Revised manuscript lines 420-423: “The detection of epithelial lineage cell marker gene expressions via RT-qPCR corroborates their physiological similarity to the native intestine. The patterns of relative expression levels for markers of enterocytes, goblet cells, and Paneth cells within the monolayers closely mirror the distribution of these differentiated cell types as observed in the human lower intestine (Hickey, et al., 2023).”

- Revised manuscript lines 444-447: “Likewise, optimum culture conditions which actively induce cellular differentiations while effectively maintaining the epithelial monolayers would be worth investigating in the future studies as it could provide an even superior in vitro model to study biological and physiological functions of the intestinal barrier as evaluated in 3D organoids (Kawasaki et al., 2023).”

Reviewer #2: 

1. Line 78 ongoing: When mentioning the collection of the tissues the authors may specify the localization of the samples, which were harvested. Were these samples harvested all at the same location or spread over a wider area of the bovine rectum?

The samples were harvested from a randomly chosen site within the same region of the rectum. To clarify this point, we modified the Materials and Methods section in the revised manuscript.

- Revised manuscript line 80: “Each animal yielded about 30 tissue pieces from a randomly selected area of the rectum, which were…”

2. Line 92: Is the medium used for the ´sterile wash´ the same washing medium as described above (PBS including Penicillin/streptomycin and Gentamicin)?

Yes. The same washing medium was used for the sterile wash. The manuscript was modified to improve clarity as below:

- Revised manuscript line 92: “… the samples underwent a sterile wash with the wash medium described above and were segregated into three equal portions…”

3. Line 95: The authors state the crypt isolation as “…described previously [4]”. Please add another phrase stating, that the isolation is described in the following.

Thank you for your suggestion. We have modified the manuscript as below:

- Revised manuscript line 95: “…for intestinal crypt isolation, which is described in the following section.”

4. Line 108: There is a typo “pelted”

The spelling was corrected to “pelleted” (Revised manuscript line 109).

5. Line 109: Are the crypts seeded after centrifugation regardless their number? It is unlikely to expect, the all harvested material contained the same amount of crypts. Since the development form crypts to organoids and the later cultivation of the organoids is highly dependent of the number of crypts/organoids per Matrigel droplet, it is mandatory to compare cultivation methods with the same amount of crypts.

We acknowledge the reviewer’s concerns about the variability in the number of crypts harvested from the samples. Indeed, there was variability in crypt numbers, with fresh tissues typically yielding a higher and more intact count of crypts compared to cryopreserved tissues, which produced significantly fewer crypts. Moreover, crypts from cryopreserved tissues often became fragile or disintegrated during the EDTA solution incubation, making them challenging to quantify accurately, as depicted in the revised Fig 1A (previously Fig 2A). Consequently, our study emphasized a “qualitative” rather than “quantitative” approach to evaluating the efficacy of the two cryopreservation methods. We defined “success” as the ability to generate organoids and maintain them in long-term culture, irrespective of the initial number of crypts isolated. This definition and approach have been clearly outlined in the Materials and Methods section of our revised manuscript to enhance understanding.

- Revised manuscript lines 127-130: “Since the number and quality of the isolated crypts varied between the tissue processing techniques, i.e. fresh or cryopreservation, the techniques were considered feasible or “success” when at least one organoid developed following the crypt isolation, consistently expanded through serial passages, and stably maintained for more than five passages.”

6. Line 112: The medium is called DMEM/F12 not DMED/F12

The spelling was corrected to “DMEM/F12” (Revised manuscript line 113).

7. Lines 134-136: How large were the Matrigel droplets, which were placed, in the 48-well plates? The same size as in the 24-well plates mentioned earlier? Why the switch in plate format?

In both the 24- and 48-well plates, we consistently applied the same volume of Matrigel, specifically 30 µL per well. The decision to transition from 24- to 48-well plates was strategically made to improve the efficiency of our culture practices in a more economical way. This change resulted in a reduction of 200 µL in the required volume of culture medium per well, while maintaining the same quantity of organoids and Matrigel, thereby offering cost savings in comparison to the use of 24-well plates. We appreciate the opportunity to clarify our methodology. To clarify these points, additional explanation was included in the Materials and Methods and Discussion sections as below:

- Revised manuscript line 140: “…. and cultured in 30 µL per well as described above in 48-well plates with 300 µL of culture medium.”

- Revised manuscript lines 356-360: “Furthermore, employing 48-well plates for all subsequent subcultures, in conjunction with the effective use of DMEM/F12 based culture medium, significantly enhanced the efficiency and cost-effectiveness of the organoid expansion process. This approach reduced the culture medium requirement by 200 µL per well, offering a notable advantage over the use of 24-well plates,”

8. Lines 154-157: One does not measure the number of molecules but the emission as a measure for the amount of transport fluorescent dye.

We appreciate your attention to an unclear expression related to our methodology in the permeability assay section. Following your observation, we have omitted the term 'molecules' and revised the manuscript to provide a more precise description of the assay as follows:

- Revised manuscript lines 159-161: “The fluorescence intensity of the culture medium in the basal chamber, which corresponds with the amount of the FITC-dextran that passed through the cell monolayer over 120 minutes, was measured …”

9. Line 166: What was the solvent of BSA?

BSA was dissolved in PBS, which was included in the manuscript for clarification (Revised manuscript line 173).

10. Line 175: Please state how the organoids were dissolved from the Matrigel in order to mount them on glass bottom dishes.

Th organoids were recovered from the Matrigel by incubating them in the 4% PFA following removal of the culture medium. A sentence describing this step was added to the Materials and Methods section to explain the technique more thoroughly.

- Revised manuscript lines 173-174: “The initial 4% paraformaldehyde treatment also allowed 3D organoids to be recovered from the Matrigel. “

11. Line 185: Please change the writing to “marker genes”

The spelling was corrected to “marker” (Revised manuscript line 194).

12. Lines 209-214: Applying a Wilcoxon test requires testing for normal distribution of the data and used when this is not the case. Please include the statement why this test is used.

Thank you for highlighting the ambiguity in the description of our statistical analysis methodology. In response to your suggestion, we have included a statement on normality assessment in the Materials and Methods section. Furthermore, after a thorough review of the statistical tests used in the original manuscript, we have made appropriate adjustments to both the Results section and the figure legends to ensure accuracy and clarity. 

- Revised manuscript lines 222-224: “The data were compared between organoids derived from fresh and slow-frozen tissues using either Wilcoxon’s signed rank test or paired t-test for independent samples upon evaluating each dataset for the normality using Shapiro-Wilk test.”

- Revised manuscript line 255 (Fig 1 legend): “Statistical analysis was performed with Wilcoxon’s signed rank tests.”

- Revised manuscript line 278 (Fig 2 legend): “No difference was noted between the two groups in all markers using Wilcoxson’s signed rank test (SOX9) or paired t-tests (SNA and EdU).”

- Revised manuscript line 298-300 (Fig 3 legend): “Statistical analysis was performed with either paired t-test (LGR5 and CHGA) or Wilcoxon’s signed rank test (LYZC, MUC2, and FABP2) for independent samples.”

13. Figure 5E: The quality of the images is poor. Please change to images with better resolution to further improve the quality of the manuscript.

The process of capturing z-stack images, which entails compiling multiple optical sections at various depths within a specimen, inherently faces resolution limitations. These limitations may lead to pixelation, particularly when images are magnified. Nonetheless, z-stack imaging is an indispensable technique for the three-dimensional visualization and analysis of specimen architecture. The quality of the z-stack images presented in our study aligns with the standards commonly accepted within the scientific community, as demonstrated by comparable research (refer to Fig 5C in Altay, et al., 2019; Fig 2G in Roodsant, et al., 2020; Fig 5C&D in Blake, et al., 2022). Despite the pixelation concerns, our images accurately provide crucial insights: they confirm the cells' monolayer arrangement, detail the apical brush border delineated by F-actin, illustrate the basolateral adherens junctions highlighted by E-cadherin, and identify a mucus-secreting goblet cell showcasing extracellular mucin release (SNA). Considering the informative value and adherence to field standards of these images and the clarity with which these features are depicted, we respectfully opt to retain the current images in their present form in the revised manuscript.

References for z-stack confocal images:

- Altay, et al. 2019: https://www.nature.com/articles/s41598-019-46497-x

- Roodsant, et al., 2020: https://www.frontiersin.org/articles/10.3389/fcimb.2020.00272/full?report=reader

- Blake, et al., 2022: https://www.frontiersin.org/articles/10.3389/fvets.2022.921160/full

14. Lines 343-346: Please remove one of the redundant sentences.

As pointed out, the second sentence was removed from the Discussion (lines 377-378).

15. Line 349: What does a 100% success rate mean? Just the generation of an indifferent number of organoids from crypts (quantity)? Or the successful generation from organoids from crypts regardless their numbers (quality).

The word “success” was used to mean the ability to generate organoids and maintain them in long-term culture, irrespective of the initial number of crypts isolated. This definition has been clearly outlined in the Materials and Methods section of our revised manuscript to enhance clarity.

- Revised manuscript lines 127-130: “Since the number and quality of the isolated crypts varied between the tissue processing techniques, i.e. fresh or cryopreservation, the techniques were considered feasible or “success” when at least one organoid developed following the crypt isolation, consistently expanded through serial passages, and stably maintained for more than five passages.”

16. Lines 396-398: If doing comparisons to other studies, please use more than one reference. Moreover, the amount of cells/cm2 is relevant, not the total number of cells.

As pointed out, the sentence was modified to be consistent with the number of references used (line 431). Additionally, in alignment with the methodologies employed in prior studies cited in our original manuscript (Sutton, et al., 2022; van der Hee, et al., 2018), our current investigation utilized 24-well culture inserts for monolayer cultures. By reporting the total number of cells per well, our approach facilitates a direct comparison with these referenced studies. Nonetheless, we have made it a point to include the size of the cell culture area (0.33 cm2/well) for enhanced clarity and whenever it was necessary for a more comprehensive understanding (Revised manuscript lines 148, 432, and 434).

References

Adler S, Pellizzer C, Paparella M, Hartung T, Bremer S. The effects of solvents on embryonic stem cell differentiation. Toxicol In Vitro. 2006 Apr;20(3):265-71. doi: 10.1016/j.tiv.2005.06.043.

Altay G, Larrañaga E, Tosi S, Barriga FM, Batlle E, Fernández-Majada V, et al. Self-organized intestinal epithelial monolayers in crypt and villus-like domains show effective barrier function. Sci Rep. 2019 Jul 12;9(1):10140. doi: 10.1038/s41598-019-46497-x.

Blake R, Jensen K, Mabbott N, Hope J, Stevens J. The Development of 3D Bovine Intestinal Organoid Derived Models to Investigate Mycobacterium Avium ssp Paratuberculosis Pathogenesis. Front Vet Sci. 2022 Jul 4;9:921160. doi: 10.3389/fvets.2022.921160.

Charavaryamath C, Fries P, Gomis S, Bell C, Doig K, Guan LL, et al. Mucosal changes in a long-term bovine intestinal segment model following removal of ingesta and microflora. Gut Microbes. 2011 May-Jun. doi: 10.4161/gmic.2.3.16483.

Coelho, TC, Chalfun-Junior A, Barreto, HG, Duarte, MS, Garcia, BO, Teixeira, PD, et al. Reference gene selection for quantitative PCR in liver, skeletal muscle, and jejunum of Bos indicus cattle. R. Bras. Zootec., 51:e20210120, 2022 doi: org/10.37496/rbz5120210120

Heumüller-Klug S, Maurer K, Tapia-Laliena MÁ, Sticht C, Christmann A, Mörz H, et al. Impact of cryopreservation on viability, gene expression and function of enteric nervous system derived neurospheres. Front Cell Dev Biol. 2023 Jun 12;11:1196472. doi: 10.3389/fcell.2023.1196472.

Hickey JW, Becker WR, Nevins SA, Horning A, Perez AE, Zhu C, et al. Organization of the human intestine at single-cell resolution. Nature. 2023 Jul;619(7970):572-584. doi: 10.1038/s41586-023-05915-x.

Inderwies T, Pfaffl MW, Meyer HH, Blum JW, Bruckmaier RM. Detection and quantification of mRNA expression of alpha- and beta-adrenergic receptor subtypes in the mammary gland of dairy cows. Domest Anim Endocrinol. 2003 Mar;24(2):123-35. doi: 10.1016/s0739-7240(02)00211-4.

Kawasaki M, Dykstra GD, McConnel CS, Burbick CR, Ambrosini YM. Adult bovine-derived small and large intestinal organoids: In vitro development and maintenance. Journal of Tissue Engineering and Regenerative Medicine, 2023, 3095002. doi: 10.1155/2023/3095002.

Koch F, Thom U, Albrecht E, Weikard R, Nolte W, Kuhla B, et al. Heat stress directly impairs gut integrity and recruits distinct immune cell populations into the bovine intestine. Proc Natl Acad Sci U S A. 2019 May 21. doi: 10.1073/pnas.1820130116.

Lee BR, Yang H, Lee SI, Haq I, Ock SA, Wi H, et al. Robust Three-Dimensional (3D) Expansion of Bovine Intestinal Organoids: An In Vitro Model as a Potential Alternative to an In Vivo System. Animals (Basel). 2021 Jul 16;11(7):2115. doi: 10.3390/ani11072115.

Liu L, Saitz-Rojas W, Smith R, Gonyar L, In JG, Kovbasnjuk O, et al. Mucus layer modeling of human colonoids during infection with enteroaggragative E. coli. Sci Rep. 2020 Jun 29;10(1):10533. doi: 10.1038/s41598-020-67104-4.

Nelli RK, Kuchipudi SV, White GA, Perez BB, Dunham SP, Chang KC. Comparative distribution of human and avian type sialic acid influenza receptors in the pig. BMC Vet Res. 2010 Jan 27;6:4. doi: 10.1186/1746-6148-6-4.

Ogaki S, Morooka M, Otera K, Kume S. A cost-effective system for differentiation of intestinal epithelium from human induced pluripotent stem cells. Sci Rep. 2015 Nov 30;5:17297. doi: 10.1038/srep17297

Ontsouka EC, Korczak B, Hammon HM, Blum JW. Real-time PCR quantification of bovine lactase mRNA: localization in the gastrointestinal tract of milk-fed calves. J Dairy Sci. 2004 Dec;87(12):4230-7. doi: 10.3168/jds.S0022-0302(04)73568-7.

Owen CD, Tailford LE, Monaco S, Šuligoj T, Vaux L, Lallement R, et al. Unravelling the specificity and mechanism of sialic acid recognition by the gut symbiont Ruminococcus gnavus. Nat Commun. 2017 Dec 19;8(1):2196. doi: 10.1038/s41467-017-02109-8.

Park KW, Yang H, Lee MG, Ock SA, Wi H, Lee P, et al. Establishment of intestinal organoids from small intestine of growing cattle (12 months old). J Anim Sci Technol. 2022 Nov;64(6):1105-1116. doi: 10.5187/jast.2022.e70.

Punyadarsaniya D, Liang CH, Winter C, Petersen H, Rautenschlein S, Hennig-Pauka I, et al. Infection of differentiated porcine airway epithelial cells by influenza virus: differential susceptibility to infection by porcine and avian viruses. PLoS One. 2011;6(12):e28429. doi: 10.1371/journal.pone.0028429.

Roodsant T, Navis M, Aknouch I, Renes IB, van Elburg RM, Pajkrt D, et al. A Human 2D Primary Organoid-Derived Epithelial Monolayer Model to Study Host-Pathogen Interaction in the Small Intestine. Front Cell Infect Microbiol. 2020 Jun 9;10:272. doi: 10.3389/fcimb.2020.00272.

Sato T, Stange DE, Ferrante M, Vries RG, Van Es JH, Van den Brink S, et al. Long-term expansion of epithelial organoids from human colon, adenoma, adenocarcinoma, and Barrett's epithelium. Gastroenterology. 2011 Nov;141(5):1762-72. doi: 10.1053/j.gastro.2011.07.050.

Shakya R, Jiménez-Meléndez A, Robertson LJ, Myrmel M. Bovine enteroids as an in vitro model for infection with bovine coronavirus. Viruses. 2023 Feb 27. doi: 10.3390/v15030635.

Sutton KM, Orr B, Hope J, Jensen SR, Vervelde L. Establishment of bovine 3D enteroid-derived 2D monolayers. Vet Res. 2022 Mar 2. doi: 10.1186/s13567-022-01033-0.

van der Hee B, Loonen LMP, Taverne N, Taverne-Thiele JJ, Smidt H, Wells JM. Optimized procedures for generating an enhanced, near physiological 2D culture system from porcine intestinal organoids. Stem Cell Res. 2018 Apr. doi: 10.1016/j.scr.2018.02.013.

---

## [Editor Report · Decision Letter 1]

11 Mar 2024

Accessible luminal interface of bovine rectal organoids generated from cryopreserved biopsy tissues

PONE-D-23-43133R1

Dear Dr. Ambrosini,

We’re pleased to inform you that your manuscript has been judged scientifically suitable for publication and will be formally accepted for publication once it meets all outstanding technical requirements.

Kind regards,

Yash Gupta, Ph.D.

Academic Editor

PLOS ONE

Additional Editor Comments (optional):

Authors have performed a commendable job in addressing the expert reviewer's comments and now the resubmitted manuscript meets the PlosOne publication standard.

---

## [Editor Report · Acceptance letter]

13 Mar 2024

PONE-D-23-43133R1 

PLOS ONE

Dear Dr. Ambrosini, 

I'm pleased to inform you that your manuscript has been deemed suitable for publication in PLOS ONE. Congratulations! Your manuscript is now being handed over to our production team.

Kind regards, 

on behalf of

Dr. Yash Gupta 

Academic Editor

PLOS ONE